# Chromophore-Targeting Precision Antimicrobial Phototherapy

**DOI:** 10.3390/cells12222664

**Published:** 2023-11-20

**Authors:** Sebastian Jusuf, Pu-Ting Dong

**Affiliations:** 1Division of Infectious Diseases, Massachusetts General Hospital, Harvard Medical School, Boston, MA 02114, USA; sjusuf@mgh.harvard.edu; 2Department of Microbiology, The Forsyth Institute, Boston, MA 02142, USA; 3Department of Oral Medicine, Infection and Immunity, Harvard School of Dental Medicine, Boston, MA 02115, USA

**Keywords:** phototherapy, endogenous chromophores, staphyloxanthin photolysis, photoinactivation of catalase

## Abstract

Phototherapy, encompassing the utilization of both natural and artificial light, has emerged as a dependable and non-invasive strategy for addressing a diverse range of illnesses, diseases, and infections. This therapeutic approach, primarily known for its efficacy in treating skin infections, such as herpes and acne lesions, involves the synergistic use of specific light wavelengths and photosensitizers, like methylene blue. Photodynamic therapy, as it is termed, relies on the generation of antimicrobial reactive oxygen species (ROS) through the interaction between light and externally applied photosensitizers. Recent research, however, has highlighted the intrinsic antimicrobial properties of light itself, marking a paradigm shift in focus from exogenous agents to the inherent photosensitivity of molecules found naturally within pathogens. Chemical analyses have identified specific organic molecular structures and systems, including protoporphyrins and conjugated C=C bonds, as pivotal components in molecular photosensitivity. Given the prevalence of these systems in organic life forms, there is an urgent need to investigate the potential impact of phototherapy on individual molecules expressed within pathogens and discern their contributions to the antimicrobial effects of light. This review delves into the recently unveiled key molecular targets of phototherapy, offering insights into their potential downstream implications and therapeutic applications. By shedding light on these fundamental molecular mechanisms, we aim to advance our understanding of phototherapy’s broader therapeutic potential and contribute to the development of innovative treatments for a wide array of microbial infections and diseases.

## 1. Introduction

Over the past few decades, the utilization of both natural and artificial light, commonly known as phototherapy, has emerged as a reliable and non-invasive approach for addressing a spectrum of illnesses, diseases, and infections. Due in part to its non-invasive nature, phototherapy is most frequently utilized to treat skin infections, such as herpes or acne lesions, where specific wavelengths of light are combined with reactive photosensitizers, such as methylene blue [1]. This type of treatment, known as photodynamic therapy, relies on the production of antimicrobial reactive oxygen species (ROS) through the photodynamic reaction between the light and an externally applied photosensitizer, thus relying on exogenous agents [2]. Recent studies, however, established the effectiveness of light alone against a wide assortment of bacterial, fungal, and viral pathogens [3]. This intrinsic antimicrobial property of light has catalyzed a shift in research focus, moving away from the photosensitivity of external agents and toward the inherent photosensitivity of specific molecules naturally found in pathogens. Chemical analyses have identified particular organic molecular structures and systems, such as porphyrins or conjugated C=C bonds, that play pivotal roles in molecular photosensitivity [4,5]. Considering the ubiquity of these systems in organic life forms, there arises a pressing need to ascertain the potential impact of phototherapy on individual molecules expressed within pathogens and discern their contributions to the antimicrobial effects of light. In this review, we delve into recently unveiled key molecular targets of phototherapy, exploring their prospective downstream implications and therapeutic applications.

## 2. Blue-Light-Photosensitive Endogenous Molecules

The term “photosensitivity” is commonly used to refer to the extent that objects react upon interacting with photons. For highly photosensitive compounds, the energy present in light is capable of being absorbed by a photosensitive molecule to reach a higher energy excited state [6]. This photoexcited molecule can react with either itself or other surrounding reagents to perform a variety of different photochemical reactions ranging from photo-dissociation, where excited molecules degrade and break down upon excitation, to photo-redox reactions, where surrounding molecules are reduced or oxidized upon contact with a photoexcited molecule [6,7]. However, the type and extent of photoreactions undertaken by photosensitive molecules, as well as the light wavelengths capable of inducing these reactions, are entirely dependent on the structure of the molecule in question. Photosensitive molecules that are specifically excited by visible light are often referred to as “chromophores” [8,9]. Over the past decade, research on the effects of antimicrobial blue light identified several endogenously expressed chromophores capable of reacting with visible blue light [10]. For this review, we focused on two types of photosensitive endogenous chromophores: pigments and porphyrins.

As compounds responsible for the phenomenon of color, pigment molecules have long been known to selectively absorb and reflect different wavelengths of light [11]. While pigments can express a wide variety of different structures and properties, recent attention has been placed on specific pigments that express a conjugated C=C bond system, which is an alternating chain of single and double bonds that was found to capture and absorb photons of various wavelengths [8]. Pigments that express this conjugated C=C bond system include carotenoids and polyenes and are responsible for the red, orange, and yellow coloration in compounds [12]. In the intricate world of microorganisms, carotenoid pigments produced by both bacterial and fungal species play a significant role in microbial metabolism and survival. Found in both bacteria and fungi, carotenoid pigments are renowned for their multifaceted contributions, including antioxidant defense against ROS, protection against heavy metal toxicity, and resilience in the face of extreme environmental temperature fluctuations [13,14,15]. The bolstered survival mechanisms afforded by microbial pigments are thus considered significant virulence factors within microorganisms, making them potential targets for therapeutic interventions [16]. 

As for porphyrins, studies on the interaction between endogenous porphyrins and light identified these compounds as key contributors to the antimicrobial effect of blue light within pathogens [17]. Unlike pigments, porphyrins contain a conjugated heterocyclic macrocycle molecular ring structure that can form complexes with numerous metallic compounds [18]. In organic life, porphyrins contribute to several metabolic processes ranging from antioxidant activity to cellular respiration by acting as a precursor to heme, which itself is a key precursor to key biological molecules, like hemoglobin, catalase, and cytochromes [19,20,21]. Within pathogens, specific endogenous porphyrins, like the heme precursors protoporphyrin IX and coproporphyrin III, were found to exhibit strong photosensitive activity and are believed to significantly contribute to the antimicrobial activity of light by reacting with oxygen to produce toxic ROS within the pathogen interior [22,23].

While photosensitive pigments and porphyrins can differ heavily in expression and structure, the common conjugated C=C bond system found in both types of endogenous photosensitive molecules has ensured that high light absorption can be found within the visible light spectrum [24]. Carotenoid pigments in general were found to experience the greatest absorption for wavelengths around 450 nm, while porphyrins intensely absorb light between the 400 and 420 nm wavelengths [25,26]. While any wavelength around these given absorbance peaks within the visible blue light spectrum can induce photoreactions, applying light close to these wavelengths with the greatest absorption will result in greater photoexcitation and reaction, therefore acting as the optimal wavelength to apply. For power settings, the following studies were conducted at powers at or below 200 mW/cm^2^, in line with ANSI safety standards. While the bulk of blue light in studies were conducted through the application of a continuous-wave light-emitting diode (LED) device, certain studies found pulsed light to have a greater efficiency in inducing photoreactions. However, both continuous and pulsed light sources remain viable methods of exciting endogenous photosensitizers. By leveraging the innate photosensitivity inherent in biological chromophores, phototherapy, specifically in the blue light spectrum, emerges as a compelling non-invasive approach to target these compounds within pathogens. This approach enhances the sensitivity to specific antimicrobial agents, promising novel avenues for treatment. 

## 3. Photolysis of Endogenous Pigments Reduces the Virulence of Corresponding Pathogens

While multiple pathogens are capable of pigment production, few hold as much medical significance as *Staphylococcus aureus* (*S. aureus*), including its more notorious antibiotic-resistant variant, methicillin-resistant *S. aureus* (MRSA). Both *S. aureus* and MRSA are notorious culprits behind a broad spectrum of medical conditions, spanning from bloodstream infections to conventional skin and soft tissue infections. Consequently, *S. aureus* infections remain a substantial source of global morbidity and mortality [27,28,29]. A distinctive feature found in over 90% of *S. aureus* is their expression of the characteristic pigment, known as staphyloxanthin (STX), which is responsible for the trademark golden yellow hue frequently used for identifying *S. aureus* strains [30,31]. In addition to its role in maintaining structural membrane stability, STX also plays a pivotal role in antioxidant defense against oxidative stress and immune cell evasion [32,33,34]. Thus, staphyloxanthin can be considered a key virulence factor for *S. aureus*, with some studies exploring the usage of cholesterol synthesis inhibitors to inhibit the synthesis of STX in *S. aureus* [35]. 

Nevertheless, owing to the conjugated C=C structure, STX exhibits an exceptionally high sensitivity to visible light (Figure 1A,B). It has been documented that this pigment displays a prominent absorbance peak at 460 nm, falling within the blue light wavelengths [36,37,38]. Comprehensive analysis employing both Raman spectroscopy and mass spectrometry unveiled that exposing isolated STX to 460 nm blue light leads to the cleavage of the conjugated C=C double bonds. This observation underscores the photolysis behavior exhibited by STX when subjected to 460 nm irradiance (Figure 1C–E). A more in-depth exploration into the specific photochemistry governing the interaction between blue light and STX revealed that the degradation of STX followed a second-order photobleaching mechanism via a triplet-triplet annihilation reaction: T* + T* → R + S (Figure 1B). In this process, light-excited STX triplet molecules (T*) react with one another, ultimately yielding a reduced (R) and semi-oxidized (S) form, causing the disintegration of the STX structure. This triplet excitation phenomenon occurs with significant efficiency in carotenoids, particularly those prone to aggregation within cell membranes [39]. Earlier studies suggested that STX molecules within the *S. aureus* membrane are organized into functional membrane microdomains (FMMs), which both structurally and functionally resemble the lipid rafts commonly found in eukaryotic cells [40]. These embedded STX FMMs not only play a pivotal role in enhancing membrane permeability and fluidity within the *S. aureus* pathogen but also serve as crucial anchor points and assembly platforms for various protein complexes responsible for the observed antibiotic resistance mechanisms in MRSA, including the penicillin-binding protein (PBP2a) [41].

The initial investigations into STX photobleaching employed 410 nm continuous-wave (CW) light-emitting diodes (LED) [37]. Subsequent studies, however, revealed that nanosecond pulsed 410 nm blue light accomplished STX photobleaching at a significantly higher efficiency, speed, and thoroughness compared with LED systems. This method demanded less than 18 J/cm^2^ of light exposure to photobleach 80% of the STX content within MRSA cells [42]. In contrast, LED systems necessitated nearly ten times that dosage, totaling 180 J/cm^2^ to attain an equivalent degree of photobleaching. This augmented efficiency is likely attributable to the microsecond-scale lifetime of the excited triplets of STX. As the high peak power present in a nanosecond pulse is able to excite the STX molecules to a triplet state within a single pulse, this yields nonlinear increments in excitation (Figure 1F). Interestingly, this increased pulsed light bleaching efficiency was not observable when applied to extracted STX resuspended within solutions. This observation underscores the dependency of STX photobleaching efficiency on concentration, emphasizing that the densely packed and aggregated configuration of STX within MRSA membranes. While both LED and pulsed light devices remain viable options for inducing STX photolysis within MRSA, the improved efficiency of pulsed light can offer a potential method to reduce and limit light exposure dosages, improving the viability of this technology within clinical environments.

The application of 460 nm blue light for photolyzing STX within *S. aureus* FMM leads to the disintegration of these domains, resulting in the formation of permeable pores measuring approximately 20~30 nm across the entirety of the bacterial membrane and subsequent cell death (Figure 1G) [42]. This pore formation not only enhances the overall membrane permeability but also disrupts the lipid packing within the domain, significantly increasing the membrane fluidity. Moreover, the disintegration of FMM was observed to trigger the detachment of anchored protein complex assemblies, including the antibiotic-resistant PBP2a protein assembly, which was found to be released from the membrane of MRSA following blue light treatment [42]. By harnessing the potential of blue-light-induced STX photobleaching to initiate pore formation and disrupt resistance proteins within the *S. aureus* membrane, multiple novel applications and therapeutic treatments involving blue light can be devised to target and combat antibiotic-resistant *S. aureus* infections.

The removal of STX from both *S. aureus* and MRSA using blue light immediately heightened their sensitivity to ROS in both in vitro and in vivo environments. The combination of blue light and low-concentration hydrogen peroxide not only synergistically eradicated *S. aureus*, biofilms, and persisters in vitro but also significantly reduced the MRSA bioburden in a murine abrasion infection model [37]. Furthermore, the depletion of STX via light-induced photobleaching can increase the killing efficiency of macrophages containing phagocytosed MRSA cells, thus enhancing the sensitivity of the immune system to MRSA. Macrophages, as key immune cells, are known to generate ROS internally to effectively eliminate internalized pathogens [16]. 

While the immediate depletion of STX increased ROS sensitivity, the pores formed during the disintegration of the STX microdomains also improved the efficacy and efficiency of antibiotics and other ROS-producing antimicrobial agents (Figure 1H). Analysis of the size of the pores was performed by measuring the intake of dextran-labeled fluorescein isothiocyanate (FITC-dextran) into light-treated MRSA cells. FD70, which is a FITC-dextran molecule with a molecular weight of 70 kDa and a Stokes radius of 6 nm, was found to be inserted in light-treated cells, indicating a minimum pore size of 12 nm (Figure 1I). Further analysis with additional FITC-dextran indicates that pores formed by light treatment have an upper limit of 30 nm. The increased membrane permeability resulting from pore formation not only improved the uptake and incorporation of membrane-targeting antibiotics, like daptomycin, but also enhanced the bacterial interior accessibility for larger antimicrobial particles, such as silver nanoparticles, thereby bolstering their overall antimicrobial efficiency and effectiveness [42,43]. One of the most significant consequences of STX photobleaching is the detachment of antibiotic resistance proteins from the MRSA membrane. This detachment allows for the resensitization of previously ineffective antibiotics, like tetracycline, ofloxacin, oxacillin, ciprofloxacin, and linezolid, against blue light-treated MRSA [42]. 

While the resensitization of conventional antibiotics against light-treated MRSA presents an intriguing avenue for circumventing established antibiotic resistance mechanisms, questions arise regarding the potential for resistance development within STX-photobleached MRSA. A 48-day serial passaging study involving MRSA subjected to approximately 90 J/cm^2^ of light revealed a near complete loss of STX expression, with colonies transitioning from a golden yellow to pure white over the course of passaging (Figure 1J) [42]. However, this depletion of STX causes drastic changes in antibiotic resistance, underscoring that the photobleaching process can ultimately eliminate a key virulence factor within MRSA. More importantly, it was observed that the addition of light treatment actually inhibited the overall resistance-developing capabilities of MRSA. In a 48-day serial passage experiment involving the daily treatment of MRSA with ciprofloxacin, it was revealed that the susceptibility of MRSA to ciprofloxacin did not change under the light treatment, with a MIC ≤ 2 μg/mL, whereas the MIC of the untreated MRSA surged to 128 μg/mL. Similar results were also noted with oxacillin. In all instances of light-treated MRSA strains within the antibiotic serial passaging studies, the depletion of STX was evident, suggesting that the efflux pumps responsible for antibiotic resistance, such as ciprofloxacin and oxacillin, are localized within the STX-rich FMM. Furthermore, this underscores that STX expression is pivotal for the localization of efflux pumps. Hence, the application of STX photobleaching not only revives conventional antibiotics against MRSA but also curbs and impedes the development of resistance to remaining susceptible antibiotics. While the ability of blue light phototherapy to enhance the effectiveness of ROS-producing antimicrobial agents is noteworthy, its capacity to rejuvenate conventional antibiotics against MRSA while mitigating the risk of resistance development presents a potentially groundbreaking and cost-effective method to treating MRSA infections.

While a significant portion of studies has primarily centered on the pigment photobleaching potential of blue light, with a focus on STX within the MRSA membrane, it is crucial to recognize that other pigment-producing pathogens have also shown susceptibility to blue-light-induced pigment photobleaching. An illustrative example is the capacity of blue light to photobleach the red-orange pigment granadaene found in the membrane of *Streptococcus agalactiae* (*S. agalactiae*), commonly known as Group B *Streptococcus* (GBS) (Figure 1K) [44,45]. *S. agalactiae* is a beta-hemolytic pathogen that has been identified as one of the leading causes of delayed wound healing and infection within cutaneous wounds, alongside *S. aureus* and *Pseudomonas aeruginosa* (*P. aeruginosa*) [46]. Like most pigments, granadaene contains a highly conjugated C=C system that renders it photosensitive to light, and like STX, granadaene has been understood to contribute to the overall survival and virulence of *S. agalactiae* [47,48]. Investigations into the role of granadaene indicated that not only does the pigment play a role as an antioxidant to shield *S. agalactiae* from oxidative damage but also the pigment was found to be responsible for the hemolytic activity heavily associated with beta-hemolytic streptococcus [49,50,51,52]. Examination of the effects of various blue light wavelengths on the granadaene pigment expressed by *S. agalactiae* demonstrated that 430 nm blue light led to the photodegradation of granadaene. Raman spectroscopy further confirmed a significant reduction in the total amount of C=C double bonds detected [53]. Beyond sensitizing *S. agalactiae* to both H_2_O_2_ and daptomycin, the photobleaching of granadaene was found to markedly diminish the hemolytic activity of *S. agalactiae* by nearly 50%, providing additional evidence of the pigment’s substantial involvement in the hemolytic activity of this pathogen [53].

Taken together, these findings collectively reinforce the feasibility of harnessing blue light to selectively target pigments within pathogens (Table 1). This approach not only diminishes bacterial defenses and virulence but also significantly enhances the effectiveness of established antimicrobial agents. Molecularly targeting pigments within pathogens stands out as a particularly pathogen-specific strategy within the realm of blue light phototherapy.

## 4. Photoinactivation of Catalase Sensitizes a Wide Range of Bacteria to Reactive Oxygen Species

Despite the benefits of pigment photobleaching, one key drawback to this technology is its selectivity. Since STX is exclusively expressed in *S. aureus* strains, STX photobleaching can only be employed to target and treat *S. aureus* infections. To expand the application of phototherapy for infection treatment, a more broad-spectrum molecular phototherapy target must be identified. While multiple ubiquitous compounds within biological organisms were discovered to be photosensitive to visible light, one molecule, in particular, stood out.

Catalase, which is a heme-containing molecule (Figure 2A), is known to be expressed in a diverse range of bacterial, fungal, and viral pathogens. Consisting of four iron-containing heme molecules, catalase is primarily responsible for efficiently neutralizing the toxic hydrogen peroxide (H_2_O_2_) to water and oxygen [54,55]. By shielding cells from oxidative stress, catalase indirectly serves as a defense mechanism against the host immune system. This capability aids catalase-positive pathogens in surviving the ROS burst produced by neutrophils and macrophages, allowing engulfed pathogens to persist in the immune cell environment [56,57,58].

Prior investigations into catalase properties had already established that the enzyme exhibited a substantial absorbance peak in the visible spectrum, specifically within the range of 405 to 410 nm blue light wavelengths [59,60]. Building upon this knowledge, research conducted by the Cheng Lab revealed that exposure to this blue light wavelength range resulted in significant disruptions in the structural and enzymatic activity of catalase (Figure 2B–D). Through Raman spectroscopy, it was found that treatment with 410 nm blue light led to a marked reduction in the characteristic peaks associated with the porphyrin heme groups (Figure 2E). Complementary catalase activity assays further demonstrated that blue light wavelengths spanning from 400 and 420 nm exerted a nearly 50% reduction in the activity of isolated bovine catalase activity (Figure 2D) [61]. Furthermore, this reduction in catalase activity was also found to be reflected within catalase-positive bacterial strains, such as MRSA and *P. aeruginosa*, following treatment with 410 nm blue light. These findings unequivocally indicated that catalase present within bacterial cells could indeed be inactivated through exposure to 410 nm blue light.

**Figure 2 cells-12-02664-f002:**
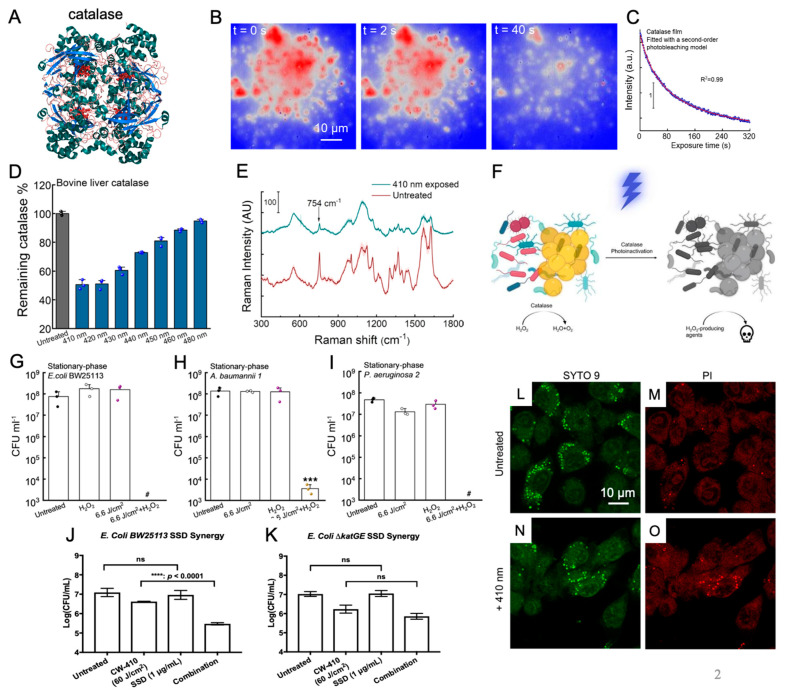
Photoinactivation of catalase sensitizes bacteria to exogenous sources of reactive oxygen species (ROS). (**A**) Molecular structure of catalase. Catalase contains four iron-containing heme groups that act as an active site for the breakdown of H_2_O_2_. (**B**) Transient absorption imaging of bovine liver catalase captured at a pump/probe wavelength of 410/520 nm through a time-course manner. (**C**) Real-time recording of catalase photoinactivation based on the signals in panel (**B**). The resulting catalase decay curve is fitted under a second-order photobleaching model. (**D**) Remaining catalase activity from bovine liver catalase following treatment with 15 J/cm^2^ of varying wavelengths of blue light. Treatment with 410 to 420 nm of light resulted in a 50% reduction in catalase activity. (**E**) Resonance Raman spectroscopy of bovine liver catalase treated with 30 J/cm^2^ of 410 nm blue light. Raman peaks corresponding to catalase (754 cm^−1^) disappear following treatment. (**F**) Schematic illustration demonstrating the increased ROS sensitization induced by catalase photoinactivation in catalase-positive bacteria strains. (**G**–**I**) Colony-forming unit (CFU) assays of various pathogens treated with 410 nm light and incubated with 22 mM of H_2_O_2_ for 30 min. (**J**) CFU assay of *E. coli* BW25113 treated with 410 nm light and silver sulfadiazine. (**K**) CFU assay of the catalase deficient *E. coli* Δ*katGE* mutant treated with 410 nm light and silver sulfadiazine. In the catalase-deficient mutant, light had no impact on silver sulfadiazine performance. (**L**–**O**) Confocal images of intracellular live (SYTO 9) and dead (PI) MRSA inside RAW264.7 macrophages without (**L**,**M**) and with (**N**,**O**) 410 nm treatment. #: below the detection limit. ****: *p* < 0.0001. ***: *p* < 0.001. ns—not significant. Panels (**A**–**I** and **L**–**O**) alongside panels (**J**,**K**) were adapted from papers [61,62] with the authors’ permission, respectively.

By combining blue-light-induced bacterial catalase deactivation with exogenous sources of ROS, such as hydrogen peroxide, it was found that 410 nm light treatment induced significant ROS sensitization among treated bacterial pathogens, creating a synergistic relationship between the photoinactivation of catalase and these ROS-producing agents (Figure 2F). In a study [61] utilizing *E. coli* BW25113 as a model bacterial strain, the incorporation of 410 nm light with a low H_2_O_2_ concentration was found to improve the antimicrobial efficiency of the compound within a time-killing assay (Figure 2G). To confirm this ROS sensitization phenomenon was due to the deactivation of catalase, identical time-killing assays were performed using the double-catalase knockout mutant *E. coli* ∆*katGE*. When comparing the two assays, the time-killing curve of the light and H_2_O_2_-treated *E. coli* wild type was found to be nearly identical to the time-killing curve of the *E. coli* ∆*katGE* mutant treated only with H_2_O_2_, underscoring the pivotal role of catalase during this ROS sensitization process. Interestingly, increased ROS sensitization was also observed in the ∆*katGE* mutant, indicating that there may be additional photosensitive molecules present within bacteria, although catalase remained the largest overall contributor to the antioxidant activity. The ROS sensitization after photoinactivation of catalase was found to stand true for multiple catalase-positive bacterial pathogen strains, including *P. aeruginosa* (Figure 2I), *S. enterica*, *A. baumannii* (Figure 2H), and *K. pneumoniae* [61]. When blue light treatment was combined with a low H_2_O_2_ concentration, these catalase-positive pathogens were found to experience near-complete eradication in the population (Figure 2G–I). In contrast, catalase-negative pathogens, like *E. faecalis*, failed to exhibit the same extent of ROS sensitization, further confirming that catalase was a primary target of blue light. These in vitro results were able to be translated to an in vivo murine model, where the antimicrobial performance of H_2_O_2_ against *P. aeruginosa*-infected abrasion wounds was significantly enhanced by light exposure.

While light-induced catalase inactivation was clearly able to sensitize catalase-positive bacterial pathogens to exogenous H_2_O_2_ from both in vitro and in vivo environments, additional experiments revealed that 410 nm light was also able to sensitize pathogens to more indirect exogenous sources of ROS, such as the ROS produced indirectly by antibiotics. While only a minor contribution to the overall mechanism of action of most antibiotics, traditional antibiotic classes, like aminoglycosides and quinolones, are understood to stimulate and activate the citric acid cycle, resulting in significant internal ROS production [63,64]. As such, photoinactivation of catalase was found to improve the performance of antibiotics, like tobramycin and ciprofloxacin, against multiple bacterial pathogens [61]. In contrast, antibiotics, like silver sulfadiazine, are known to be able to produce ROS by disrupting the bacterial respiratory chain through protein disruption from silver ions [65,66,67]. Investigation into the potential synergy between photoinactivation of catalase and silver sulfadiazine revealed that catalase depletion results in significantly increased ROS production by silver sulfadiazine, significantly improving the antimicrobial efficiency of silver sulfadiazine against both *P. aeruginosa* and MRSA. The synergy between catalase photoinactivation and silver sulfadiazine was confirmed and validated with a checkerboard assay against a model bacterium in the form of *E. coli*. Through this assay, the individual and combined minimum inhibitory concentrations (MICs) of two different treatments can be found to calculate the overall fractional inhibitory concentration index (FICI). The FICI can be used to identify the type of relationship between two agents, with synergistic interactions defined by FICI values ≤ 0.5 [68]. Based on the combined and individual MIC values of 410 nm light treatment and silver sulfadiazine obtained through the checkerboard assay, the FICI of light treatment and silver sulfadiazine was found to be 0.375, indicating that the relationship between light and silver sulfadiazine is synergistic. Additional tests on *E. coli* found that while the addition of light treatment improved the performance of wild-type *E. coli* BW25113 (Figure 2J), the same light treatment had no impact on the antimicrobial activity of catalase deficient *E. coli* Δ*katGE* (Figure 2K), indicating that the inactivation of catalase is responsible for the improved silver sulfadiazine performance. Overall, this synergistic relationship between catalase photoinactivation and silver sulfadiazine was also found to be reflected within an in vivo murine abrasion model, demonstrating the viability and potential translatability of this combination therapy for the treatment of infected wounds [62].

Although antibiotics represent just one potential exogenous source of ROS capable of synergizing with light-induced catalase inactivation, another ROS source is inherent to the host immune system. While antibiotics generate ROS internally within bacterial pathogens, immune cells, such as macrophages, employ an external ROS exposure method through their “ROS burst”. This burst is generated within macrophages via NADPH oxidase (NOX2) proteins [69], allowing them to eliminate ingested pathogens. Catalase was found to defend pathogens, like *S. aureus* and *Campylobacter jejuni*, from the ROS burst produced by macrophages during phagocytosis, demonstrating the protective role catalase plays in bacterial survival [59,70]. To determine whether photoinactivation of catalase can sensitize pathogens to the ROS burst produced by MRSA, light-treated MRSA was allowed to be phagocytized by RAW264.7 macrophages. Following phagocytosis, the bacteria and macrophage co-culture were allowed to incubate for 1 h before live (SYTO 9)/dead (PI) staining was performed on the phagocytosed MRSA (Figure 2L–O). Confocal imaging revealed that 410 nm light treatment significantly increased the amount of dead MRSA within macrophages, indicating that photoinactivation of catalase improved the killing efficiency of the macrophages. The improved killing efficiency of macrophages against light-treated pathogens was further validated based on CFU measurements of the surviving phagocytosed bacteria. To confirm that this improved killing efficiency was due to the synergy between the ROS burst and the deactivation of catalase, a NOX inhibitor known as diphenyleneiodonium chloride (DPI) was used to neutralize the ROS burst within macrophages. The addition of DPI was found to neutralize the improved killing efficiency of macrophages with light-treated pathogens, confirming that the photoinactivation of catalase is indeed synergizing with the ROS burst produced by the immune system [61].

Collectively, these studies shed light on catalase as a potential broad-spectrum target for blue light phototherapy, offering the prospect of enhanced ROS sensitization across various catalase-positive pathogens (Table 2). This increased ROS sensitization not only bolsters the antimicrobial efficacy of direct ROS sources, such as H_2_O_2_, in both in vitro and in vivo settings but also elevates the performance of conventional antibiotics like ciprofloxacin and silver sulfadiazine. Additionally, light-induced ROS sensitization was shown to enhance the killing efficiency of macrophages by augmenting the effectiveness of the ROS burst, hinting at the promising applicability of photoinactivation of catalase in wound environments. 

## 5. Photoinactivation of Catalase Sensitizes *Candida* spp. and *Candida auris* to Reactive Oxygen Species

While the catalase-deactivating capabilities of 410 nm blue light offer new and exciting opportunities for the treatment of bacterial infections, it is important to note that drug-resistant bacterial pathogens are not the only rising concern within the global healthcare community. Over the past few decades, drug resistance has developed within invasive fungal pathogens, like *Candida* spp., posing a severe health risk to immunocompromised patients [71,72,73]. While *Candida albicans* continues to be the predominant cause of invasive *Candida* infections globally, recent trends witnessed the emergence and rapid spread of *Candida* strains that are resistant to multiple drugs, particularly notable in the case of *Candida auris* [74,75]. This phenomenon can, in part, be attributed to the impact of the COVID-19 pandemic, which led to an increase in hospitalizations and compromised immune systems among those affected by the viral infection. Consequently, several *Candida* outbreaks, including those involving *C. auris*, occurred in hospital settings. These events underscore the pressing need for the development of novel approaches to diagnose, treat, and combat these infections [75]. As eukaryotic organisms, fungal strains, such as *Candida*, are recognized for their expression of catalase, which plays a crucial role in their antioxidant defense system. These fungi employ this enzyme to counteract oxidative stress triggered by the immune system, mirroring the mechanisms seen in bacterial organisms [76,77]. Based on this similarity, the impact of photoinactivation of catalase through blue light exposure was explored on various *Candida* fungal strains.

**Figure 3 cells-12-02664-f003:**
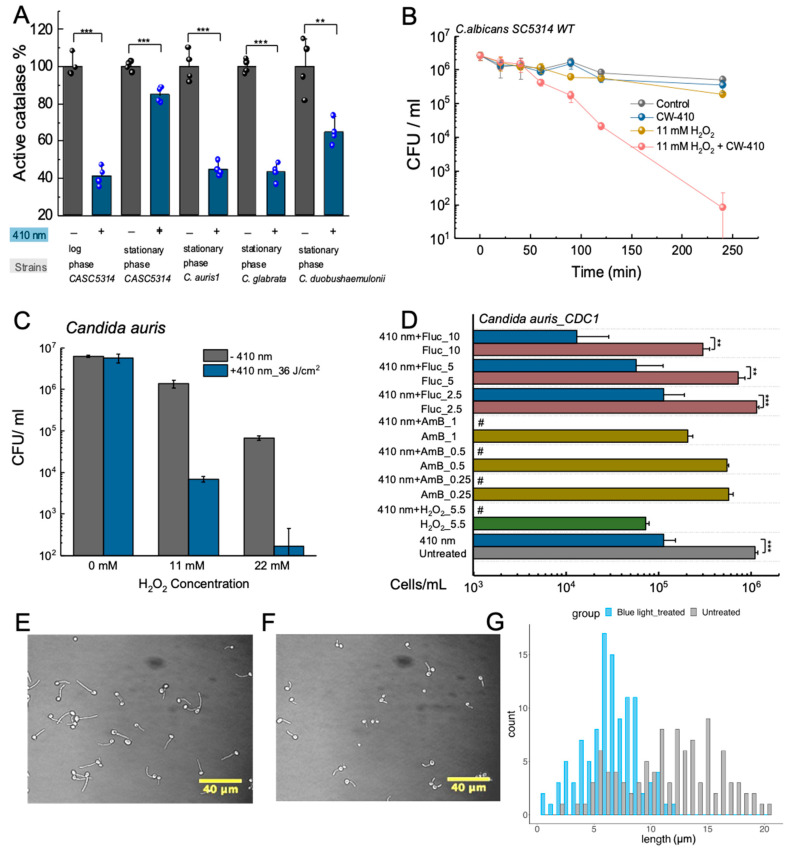
Catalase photoinactivation sensitizes fungal pathogens to exogenous sources of ROS and suppresses *Candida* hyphae development. (**A**) Remaining catalase activity from various *Candida* fungal species following treatment with 15 J/cm^2^ of 410 nm blue light. (**B**) Time-killing assay of wild-type *C. albicans* SC5314 treated with 30 J/cm^2^ of 410 nm blue light incubated alongside 11 mM of H_2_O_2_ in yeast extract-peptone-dextrose (YPD) broth. CFU/mL was quantified over the course of 4 h. (**C**) CFU/mL assay of *C. auris 1* strain treated with 36 J/cm^2^ of 410 nm light and various concentrations of H_2_O_2_ for 4 h. The addition of light significantly improved H_2_O_2_ activity against *C. auris*. (**D**) CFU/mL values of *C. auris 1* strain derived from a PrestoBlue proliferation assay and calibration curve. *C. auris* was treated with 30 J/cm^2^ of blue light and incubated with varying concentrations of amphotericin B or fluconazole. (**E**) Phase contrast imaging of untreated *C. albicans* SC5314 following 1 h incubation under hyphae-forming conditions. (**F**) Phase contrast imaging of *C. albicans* SC5314 treated with 60 J/cm^2^ of 410 nm light following 1 h incubation under hyphae-forming conditions. (**G**) Histogram of untreated versus blue-light-treated *C. albicans* hyphae. Light treatment significantly reduces average hyphae length. #: below the detection limit. **: *p* < 0.01; ***: *p* < 0.001. Panels (**A**–**G**) were adapted from papers [78,79] with the respective authors’ permission.

Research conducted on multiple *Candida* strains, including *C. auris* and *C. glabrata*, found that exposure to 410 nm blue light results in a nearly 60% reduction in catalase activity, demonstrating that the fungal catalase behaves similarly to bacterial catalase and that a greater eukaryotic complexity of fungal cells has little impact on its photosensitivity (Figure 3A) [78]. Based on this reduction in catalase activity, the ROS-sensitizing effects of catalase photoinactivation were quantified within a wildtype strain *C. albicans* SC5314. While individual treatments of 410 nm light and low concentration dosages of H_2_O_2_ barely had an impact on the viability of *C. albicans*, the combination of the two resulted in significant reductions in overall fungal viability, reducing the pathogen population by nearly 4-log_10_ in the span of 4 h (Figure 3B). This increased ROS sensitization was confirmed to be attributable to the photoinactivation of catalase based on the similar ROS sensitization phenomenon observed in *C. albicans* treated with the chemical catalase inhibitor 3-Amino-1,2,4-triazole [78]. While these results confirmed that the photoinactivation of catalase achieved synergistic eradication of *Candida* fungi when combined with H_2_O_2_, additional exploration was conducted to evaluate the potential improved efficiency that could be applied to established ROS-generating anti-fungal agents, like amphotericin B. Amphotericin B was found to universally induce the accumulation of ROS species within multiple pathogenic yeast species, and increased catalase expression was found to play role in amphotericin B resistance in *Candida* strains [80,81]. Through viability assays, photoinactivation of catalase was found to improve the antifungal performance of amphotericin B against *C. albicans* strains, confirming the ability for the photoinactivation of catalase to synergize with conventional antifungal agents. Similar improvements in performance were also observed for other conventional antifungal agents, like micafungin and miconazole.

While *C. albicans* remains the most common pathogen responsible for nosocomial infections, recent developments saw the growing emergence of additional invasive *Candida* strains, such as *C. glabrata* and *C. tropicalis* [82]. Among these differing resistant *Candida* strains, *C. auris* was noted for its high mortality and morbidity, as well as its ability to survive and persist on various surfaces. Furthermore, *C. auris* was found to exhibit strong resistance to multiple antifungal classes, ranging from azoles, like fluconazole, to polyenes, like amphotericin B [83]. Thus, with the spread and growing rates of *C. auris* infections occurring worldwide [84,85,86,87], we closely examined the potential antimicrobial effect of photoinactivation of catalase on this multi-drug resistant pathogen. Examination of the potential synergy between blue light and a low H_2_O_2_ concentration found that the addition of light-induced catalase inactivation significantly improved the performance of just 22 mM of H_2_O_2_ by nearly 3-log_10_ (Figure 3C) [78]. Additional strains of *C. auris* were found to exhibit even better improvements in fungal clearance. These results indicate that *C. auris* strains remain susceptible to the ROS-producing agents under blue light irradiance. This ROS-sensitizing effect can be used in conjunction with existing antifungal agents that are known to produce ROS as part of their mechanisms of action, such as fluconazole [88] and amphotericin B [89]. Through high-throughput PrestoBlue viability assays and CFU quantification, blue light was found to significantly improve the antifungal capability of both amphotericin B and fluconazole against proliferating *C. auris*, indicating that photoinactivation of catalase can provide a non-invasive and non-drug reliant method of bypassing established resistance (Figure 3D).

Following the in vitro exploration of the potential therapeutic synergy between the photoinactivation of catalase and antifungal agents, the potential translatability of applying light-induced catalase photoinactivation on fungal skin infections was determined through a combination of in vitro immune cell assays, as well as in vivo murine models. Like the bacterial tests, *C. albicans* fungi treated with blue light were found to boost macrophage killing of *C. albicans*, and within *C. albicans*-infected murine abrasion wounds, treatment with a combination of 410 nm light and 0.5% H_2_O_2_ was found to significantly improve the performance of H_2_O_2_ in reducing the fungal bioburden while inducing no detectable damage to the murine skin [78]. However, one interesting observation found among these results was that blue light treatment appeared to inhibit or suppress the development of *C. albicans* hyphae. As a dimorphic fungi, *C. albicans* is unique in that it can grow and reproduce through both unicellular yeast budding and filamentous hyphae formation [90]. This dimorphism gives *C. albicans* strong survival and virulence capabilities, providing the fungi with a method of escaping macrophage phagocytosis through the mechanical force exerted by hyphae or providing a method of improving invasion capabilities through the epithelial cells [91,92]. Previous studies established a close link between *Candida* catalase expression and hyphae growth, as *C. albicans* with double-catalase gene disruptions exhibited suppressed hyphae growth [93]. Based on these previous observations and studies, the impact of catalase photoinactivation on hyphae growth was closely examined.

While untreated *C. albicans* grown in hyphae-inducing conditions were found to express strong hyphae growth (Figure 3E), *C. albicans* treated with 60 J/cm^2^ of 410 nm light exhibited inhibited hyphae growth (Figure 3F) [79]. Measurements determined that while the untreated hyphae had an average hyphae length of 11.62 ± 4.22 μm, the light-treated hyphae were significantly shorter at 6.47 ± 2.39 μm, corresponding to an average hyphae length reduction of 44.3%. Further histogram analysis revealed that while over 72% of untreated hyphae were over 10 μm in length, only 11% of light-treated hyphae reached that threshold, indicating a clear suppression in overall hyphae development (Figure 3G). Further study into the potential metabolic changes induced by light-induced catalase photoinactivation found the light treatment to significantly reduce the metabolic activity of *C. albicans*, with a specific apparent disruption in lipid droplet formation, indicating a potential alteration in lipid metabolism caused by light treatment. The usage of gas chromatography-mass spectroscopy (GC-MS) found significant reductions in both unsaturated and saturated fatty acid production within the light-treated *C. albicans*, indicating that the deactivation of catalase appears to cause a significant disruption in lipid metabolism, therefore suppressing hyphae development. These results line up with previously established literature, which indicates that disruption in sterol or sphingolipid biosynthesis results in impaired hyphae development [94].

Taken together, light-induced catalase deactivation not only sensitizes fungal *Candida* strains (Table 3) to ROS sources like H_2_O_2_ and amphotericin B, but the deactivation of catalase is capable of significantly reducing the survival and invasion capabilities of *C. albicans* by inhibiting hyphae formation through the disruption of lipid metabolism. Coupled with the fact that this light treatment remains effective against even multi-drug-resistant *Candida* pathogens, like *C. auris*, light-induced catalase deactivation offers a non-invasive and easily translatable method of targeting and treating *Candida* surface infections. 

**Table 3 cells-12-02664-t003:** Summary of the photosensitivity of fungal catalase, its function within biological pathogens, and the effects of photosensitization.

Chromophore	Pathogen Expression	Peak Absorbance Wavelength	Biological Function	Effects of Photoexcitation
Fungal Catalase	Catalase-positive fungi (*C. albicans*, *C. auris*, *C. tropicalis*, *C. glabrata*)	405 nm [60]	Antioxidant defense [76], immune cell evasion [77], and hyphae formation and growth [93]	Increased ROS sensitivity, increased antibiotic susceptibility [78], and reduced hyphae formation and growth [79]

**Figure 4 cells-12-02664-f004:**
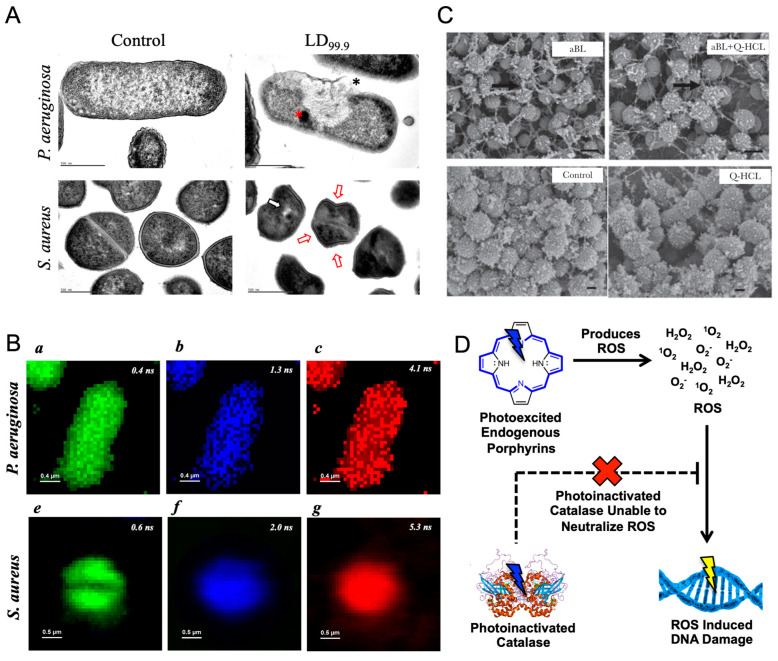
Characterization of morphology and ultrastructure of the bacteria and bacterial biofilms in response to reactive oxygen species (ROS) generated from endogenous porphyrins and catalase under blue light irradiance. (**A**) Representative images from transmission electron microscopy illustrating aBL-inducing ultrastructural damage in *P. aeruginosa* and *S. aureus*. Red asterisk, agglutination of intracellular contents; black asterisk, cell wall/membrane damage; white arrow, leakage of intracellular contents; red arrow, membrane destabilization. Scale bar: 500 nm. Abbreviations: LD_90_ and LD_99.9_, lethal doses responsible for 90% and 99.9% killing, respectively. (**B**) Fluorescence lifetime imaging (FLIM) images of endogenous fluorophores from *P. aeruginosa* (a to c) and *S. aureus* (e to g). Different colors represent different fluorescence lifetimes for each bacterial species (green, blue, red). (**C**) The SEM images show the morphology and ultrastructure of the bacterial biofilms after various treatments. Treatments: aBL + quinine, aBL, Quinine, or untreated control. Black arrows: biofilm matrix. Scale bar: 500 nm. (**D**) Schematic summary of the interactions between photons and two primary intrinsic photosensitive chromophores within microbes. Panels (**A**–**C**) were adapted from papers [95,96] with the respective authors’ permission.

## 6. Photoexcitation of Endogenous Porphyrins Induces Intracellular ROS and Triggers Cellular Damage

Niels Ryberg Finsen was awarded the Nobel Prize for his contribution to utilizing concentrated ultraviolet light to treat *Mycobacterium tuberculosis*-induced lupus vulgaris in 1903 [97]. This is a great application of photobiomodulation, or more specifically, photonic antimicrobial therapy. However, the underlying mechanism of photobiomodulation, especially the effect of photons on microbes, has remained elusive for decades. Here, we demonstrated that these endogenous pigments (staphyloxanthin, granadaene, catalase) are vital targets of visible lights, especially from 400 nm to 450 nm. However, we concur that these pigments do not constitute the sole factors contributing to the antimicrobial effects induced by blue light. Another widely accepted hypothesis is the cytotoxic reactive oxygen species generated by the photoexcitation of endogenous porphyrins and flavins [10,98]. 

Carolina dos Anjos et al. examined the morphology change of individual *P. aeruginosa* and *S. aureus* using transmission electron microscopy under blue light irradiance (Figure 4A) [95]. A dense cytoplasm was observed, which echoes with the cytoplasmic condensation in the presence of antibiotic lethality [99]. Meanwhile, aggregation of the cellular content and detached membrane was found. Collectively, this suggested blue light exposure elicits oxidative damage. They also quantified the concentration of intracellular heme precursor porphyrins from those bacteria, and it turned out that coproporphyrins alongside protoporphyrin IX were among the most abundant intracellular porphyrins among all the tested bacteria. These results are consistent with previous studies conducted by both Kennedy et al. and Walter et al., which found that both endogenously produced protoporphyrin IX and coproporphyrin III induced significant antimicrobial activity when combined with light [22,23]. Dos Anjos et al. also captured the potential intrinsic fluorophores at the single-cell level through fluorescence lifetime imaging (Figure 4B), which is a method that allows for achieving the spatial localization of endogenous fluorophores based on the fluorescence lifetime [100]. They found more than one certain fluorophore exists in the bacteria, which indicates the multifaceted targets of antimicrobial blue lights. 

Besides the sole antimicrobial treatment from blue light, Leon Leanse et al. also combined antimicrobial blue light with other antimicrobial adjuvants, such as quinine, to eliminate both molds and Gram-negative bacteria [96,101]. Blue light irradiance demonstrated the enhanced uptake of quinine, an antiseptic agent, at the single-cell level through label-free Raman imaging of accumulated quinine. Moreover, through the combinatorial treatment between blue light and quinine, the extracellular polymeric substances were significantly reduced and degraded in the *P. aeruginosa* biofilms (Figure 4C). And they also quantified the amount of intracellular porphyrins and found a substantial accumulation of coproporphyrins from most of the tested molds. It is well known that protoporphyrins can produce reactive oxygen species due to the photodynamic reaction under blue light irradiance [102]. Therefore, akin to the aforementioned pigments, porphyrins are regarded as another class of targets for blue light (Table 4).

## 7. Future Directions

Through various studies, the photobleaching capabilities of blue light against specific and broad-spectrum molecular targets were explored. Bleaching pigments expressed by specific pathogens, such as staphloxanthin by MRSA and granadaene by *S. agalactiae*, offers a method of disrupting the bacterial membrane of pathogens, while photoinactivation of catalase provides a universal method of sensitizing both catalase-positive bacterial and fungal strains to exogenous ROS sources. Given the importance that pigments like staphyloxanthin play in MRSA virulence and survival, some studies began examining the potential viability of staphyloxanthin biosynthesis inhibitors, like flavonoids and thymol, as a method of eliminating MRSA infections [35,103,104]. Despite this, light treatment remains advantageous in its non-invasive and non-drug-reliant nature. However, it is important to note that the molecular targets discussed within the manuscript are not the only potential targets of phototherapy. Additional colored pigments known to significantly contribute to bacterial virulence, such as violacein in *Chromobacterium violaceum* and pycocyanin in *P. aeruginosa*, all offer potential targets for phototherapy as a method of reducing bacterial virulence [16,105,106]. For more broad-spectrum targets, time-killing assays of catalase-deficient *E. coli* mutants exhibit increased sensitivity to H_2_O_2_, indicating that additional antioxidant enzymes or peroxidases can be potentially deactivated by light exposure. Thus, additional antioxidant peroxidases that likely contain porphyrins are excellent candidates to explore in terms of their photosensitivity. Recent reports also indicated that cytochrome c oxidase, a key respiration enzyme that plays a role in maintaining the proton gradient necessary for ATP production, exhibits strong photosensitivity to blue light, providing yet another potential target to explore in terms of future phototherapy studies [107,108]. Given the vast number of enzymes that incorporate heme and porphyrin molecules, there are a multitude of potential photosensitive targets to explore. While current research on endogenous photodynamic therapy remains limited to the visible blue light spectrum, the identification of additional molecular targets photosensitive at higher wavelengths remains an additional potential direction to develop this phototherapy research. The utilization of higher wavelengths would allow for the potential combination of endogenous photodynamic therapy with photothermal therapy, which primarily operates within the near-infrared region (750–900 nm) [109]. Certain endogenous pigments, like melanin, were already demonstrated to experience a photothermal reaction upon exposure to 808 nm light, and the identification of additional photoreactive molecules at higher wavelengths could allow for the development of a combined treatment that takes advantage of both the thermal and reactive reactions of light therapy [110]. 

## 8. Conclusions

In this subsequent review, we undertook an examination of recent advancements within the realm of phototherapy, with a particular focus on the specific molecular targets that display photosensitivity to light exposure in isolation. Notably, the photodegradation of membrane pigments, as exemplified by the photobleaching of staphyloxanthin in *S. aureus* using 460 nm light, presents a precise approach for targeting pigment-expressing pathogens. This approach elevates the sensitivity of reactive oxygen species (ROS) and disrupts microbial membrane integrity, consequently diminishing their antimicrobial-resistance capabilities. Concurrently, the photoinactivation of enzymes, such as catalase, through the application of 410 nm light offers a broader-spectrum strategy for enhancing ROS sensitivity and rendering catalase-positive bacteria and fungi more susceptible to immune cells. Furthermore, the intrinsic protoporphyrins expressed in bacterial strains during natural metabolism can act as endogenous photosensitizers, instigating the generation of antimicrobial ROS within the microbial interior. The utilization of light, whether for the degradation or excitation of these molecular targets, furnishes a non-invasive approach to combat pathogens. This exploits the inherent photosensitivity of virulence agents themselves, thereby reducing the likelihood of resistance development. It is crucial to underscore that this review merely marks the commencement of endeavors in the realm of chromophore-targeting precision phototherapy. As we delve deeper into exploring and identifying potential photosensitive targets, antimicrobial light therapy has the potential to become a formidable tool in the battle against antimicrobial-resistant pathogens.

## Figures and Tables

**Figure 1 cells-12-02664-f001:**
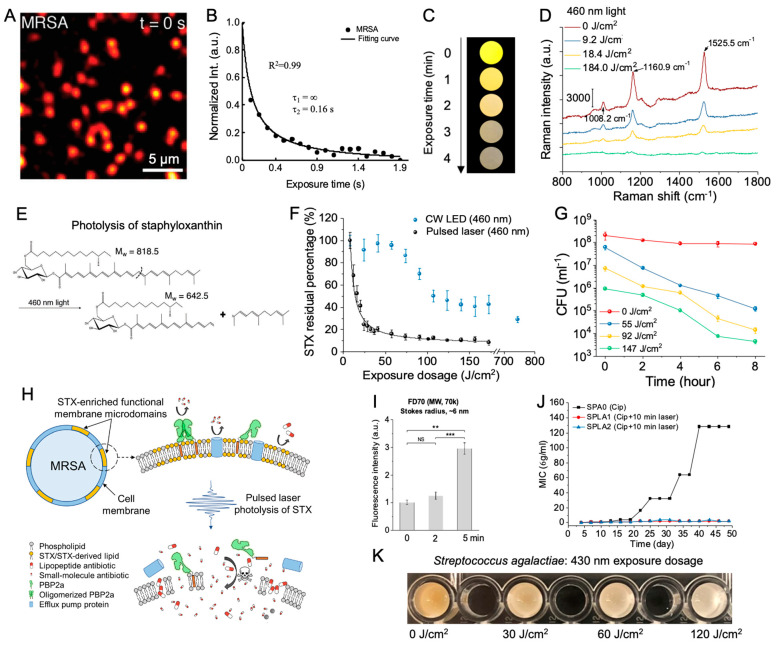
Characterization of the mechanisms and antimicrobial applications of pigment photolysis. (**A**) Transient absorption imaging of MRSA acquired at a pump/probe wavelength of 520/780 nm detected a strong initial signal associated with the staphyloxanthin (STX) chromophore at the initial time t = 0 s. The signal decayed within a second of exposure. (**B**) Representative time-lapse STX signal in MRSA. MRSA intensity was fitted to a second-order photobleaching model. (**C**) Digital images of concentrated MRSA droplets under the treatment of 460 nm blue light. Over the course of 4 min of light exposure, the golden yellow coloration in MRSA fades. (**D**) Resonance Raman spectroscopy of MRSA samples treated with pulsed blue light. Raman peak positions (labeled by wavenumber) associated with STX decrease with light exposure. (**E**) Theorized structural breakdown of STX following exposure to 460 nm light. (**F**) Comparison of the STX photolysis kinetics within MRSA treated under either continuous-wave light-emitting diode (LED) or a nanosecond pulsed laser at 460 nm under the same power conditions. The black curve on the pulsed data represents the fitting result under a second-order photobleaching model. (**G**) Time-killing assay of MRSA resuspended in phosphate-buffered saline (PBS) for up to 8 h following exposure to varying dosages of 460 nm light. (**H**) Schematic of the mechanisms behind the resensitization of conventional antibiotics in MRSA following treatment with pulsed blue light. Pore formation induced by STX photolysis of membrane microdomains disrupts pre-existing resistance mechanisms. (**I**) Quantitation of uptake of FD70 in MRSA by fluorescence following pulsed light treatment. (**J**) Characterization of resistance development of untreated and light-treated MRSA over the course of a 48-day serial passage in the presence of sub-minimum inhibitory concentration (MIC) levels of ciprofloxacin. No resistance development occurred with light-treated MRSA. (**K**) Digital image profile of concentrated *S. agalactiae* following exposure to 120 J/cm^2^ of pulsed 430 nm blue light. Over the course of light treatment, the orange associated with the granadaene pigments fades to white. ***: *p* < 0.001. **: *p* < 0.01. ns—not significant. Panels (**A**–**K**) were adapted from papers [37,42] with the respective authors’ permission.

**Table 1 cells-12-02664-t001:** Summary of photosensitive endogenous pigments, their biological function, and the effects of photoexcitation on pathogens that express them.

Chromophore	Pathogen Expression	Peak Absorbance Wavelength	Biological Function	Effects of Photoexcitation
Staphyloxanthin	*Staphylococcus aureus* [30]	460 nm [38]	Membrane stability and fluidity [32], antioxidant defense [33], and immune cell evasion [34]	Increased membrane permeability, increased ROS sensitivity [37,43], increased antibiotic susceptibility, and inhibited resistance capabilities and development [42]
Granadaene	*Streptococcus agalactiae* [44]	425 nm [45]	Antioxidant defense [50], immune cell evasion [51], and hemolytic activity [52]	Increased membrane fluidity, increased ROS sensitivity, increased antibiotic susceptibility, and reduced hemolytic activity [53]

**Table 2 cells-12-02664-t002:** Summary of the photosensitivity of bacterial catalase, its function within biological pathogens, and the effects of photosensitization.

Chromophore	Pathogen Expression	Peak Absorbance Wavelength	Biological Function	Effects of Photoexcitation
Bacterial Catalase	Catalase-positive bacteria (*E. Coli*, *S. aureus*, *P. aeruginosa*, *A. baumannii*)	405 nm [60]	Antioxidant defense [56] and immune cell evasion [57]	Increased ROS sensitivity [61], increased immune cell susceptibility, and antibiotic susceptibility [61,62]

**Table 4 cells-12-02664-t004:** Summary of the photosensitivity of porphyrins, their function within biological pathogens, and the effects of photosensitization.

Chromophore	Peak Absorbance Wavelength	Biological Function	Effects of Photoexcitation
Protoporphyrin IX	406 nm [26]	Heme precursor [21]	Increased ROS formation [22,102] and membrane damage [95]
Coproporphyrin III	395 nm [23]	Heme precursor [21]	Increased ROS formation [23] and membrane damage [95]

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
