# Peer review of "Chromophore-Targeting Precision Antimicrobial Phototherapy"

_cells, 2023, doi:10.3390/cells12222664_

Round 1
Reviewer 1 Report
Comments and Suggestions for Authors
The manuscript "Chromophore Targeting Precision Phototherapy" has an interesting approach towards antimicrobial substances.
The authors present an extremely interesting topic, which is not yet completely explored. The future directions are promising, especially now in the era of antibiotic resistance.
The manuscript lacks abstract, as well as Key words and is not following the template provided by cells. These concerns have to be adressed immediately.
Few questions remain:
1. How will be the actual in-vivo application of blue light ? It is clear, that options exists for an easy wound therapy (on the skin) or even in the oral cavity as well as in the nasal cavity could be possible to eliminate the pathogens. Can the authors describe possible applications ?
2. How could be the use inside the body, when there is a need for antimicrobial therapy ?
3. Is the laser pulse not dangerous on living objects, skin etc ?
4. Is the blue light application safe for living tissues and organs ? There was a description about research on murine? Can this be possible for other living organisms, including humans ?
5. The authors mention in one example, that there is a 30 min period of blue light usage, which reduced the pathogens. Is this in general, or there are differences for every species, and even on biofilms ?
6. The authors shortly explain about the effects on blue light on C=C bonds within STX, granadene and catalase. In general, any other C=C could be targeted by blue light within other compounds in the area of interest?
Best of Luck
Reviewer 2 Report
Comments and Suggestions for Authors
This review is very interesting, addressing the main natural pigments localized mainly in bacteria and fungi, that act as chromophores when exposed to specific light wavelengths. The text is very detailed, and well-organized, showing some illustrations adapted from other studies. I think that the contribution of the review is undoubted and very important for the research of diverse fields, including photodynamic therapy. Only minor doubts are present:
1) The title of the review did not mention that the targets are localized in microbes. I suggest a reformulation of the title, including some words pointing to this aspect.
2) Although it is clear that the Figures inserted in the manuscript are not plagiarism from the other studies (even if the authors are the same as the current manuscript), I suggest the creation of new schemas and illustrations for the current review, using the results previously published.
Reviewer 3 Report
Comments and Suggestions for Authors
This review talks about the use of phototherapy in a wide array of microbial infections and diseases, focusing especially on the intrinsic antimicrobial properties of light itself. It aims at the photosensitivity of molecules naturally present within pathogens, talking about photolysis, photoinactivation, and photoexcitation. In my opinion, the review is well-written and structured. However, there are still some topics that need to be added.
1. Because each subtopic is vast and has a lot of information, I would like to see a table at the end of each subtopic, where some of the main information is resumed.
2. Similarly, I think this review would benefit if the author added a specific topic talking only about the potential photosensitive targets, as the information may be lost in the other subtopics (e.g., types of endogenous photosensitizers, their biological roles)
3. Similarly, I think the authors should talk a bit more about the light used in this type of therapy, and some of its characteristics (wavelength selection, optimal wavelengths for specific chromophores, and light parameters).
4. The authors should talk about the combination of “endogenous” photodynamic therapy and “endogenous” photothermal therapy in the future perspectives.
Round 2
Reviewer 1 Report
Comments and Suggestions for Authors
The authors did address the concerns properly and with evidences.
The outcome is now more organized and gives better insight.
The manuscript is ready to be published.
We hope to see more research in this direction.
Congratulations